# Challenges and Opportunities in Industry 4.0 for Mechatronics, Artificial Intelligence and Cybernetics

**Vasiliki Liagkou** [1], **Chrysostomos Stylios** [1,*], **Lamprini Pappa** [1] and **Alexander Petunin** [2]

1 Department of Informatics and Telecommunications, University of Ioannina, 47150 Arta, Greece; liagkou@uoi.gr (V.L.); pappa.l@kic.uoi.gr (L.P.)
2 Institute of Mechanics and Machine-Building, Ural Federal University Named after First President of Russia B. N. Yeltsin, Mira Str., 19, 620002 Yekaterinburg, Russia; a.a.petunin@urfu.ru
* Correspondence: stylios@uoi.gr; Tel.: +30-2681050330

**Abstract:** Industry 4.0 has risen as an integrated digital manufacturing environment, and it has created a novel research perspective that has thrust research to interdisciplinarity and exploitation of ICT advances. This work presents and discusses the main aspects of Industry 4.0 and how intelligence can be embedded in manufacturing to create the smart factory. It briefly describes the main components of Industry 4.0, and it focuses on the security challenges that the fully interconnected ecosystem of Industry 4.0 has to meet and the threats for each component. Preserving security has a crucial role in Industry 4.0, and it is vital for its existence, so the main research directions on how to ensure the confidentiality and integrity of the information shared among the Industry 4.0 components are presented. Another view is in light of the security issues that come as a result of enabling new technologies.

**Keywords:** Industry 4.0; mechatronics; artificial intelligence; cybersecurity; security threats; internet of things; cyber-physical systems



## 1. Introduction

We have witnessed a significant switchover in the industry led by the global market demands in the past decade. Conventional manufacturing methods are becoming obsolete, and there is a need for more dynamic manufacturing processes. New manufacturing operations have been introduced, and effective factory management has become of great interest [1]. In order to address the needs of modern manufacturing, high-tech elements have been utilized, such as sensors, software, and wireless connectivity [2]. To sum up, there has been a transition from digital to intelligent systems in the manufacturing sector that emerged due to the rapid progress in electrics, electronics, information, and advanced manufacturing technology [3].

The 4th industrial revolution, known by the abbreviation I4.0 (i.e., Industry 4.0), was first announced in Germany in 2011. An ambitious high-tech governmental project shook to the core and ultimately transformed the entire production cycle. This new era is characterized by computers and automation, leading to new terms in the industrial sector, such as smart automation, self-optimization, self-configuration, and self-diagnosis/prognosis. Principles of machine learning (ML) and artificial intelligence (AI) are widely applied, such as to robotic systems, which are remotely connected through the internet to cloud servers in order to gain the ability to learn and control them with the potentially most minor intervention of human operators [4]. The critical factor of I4.0 is that it embodies the collection and analysis of real-time data for the sake of productivity increase [5]. Within this industrial revolution, rather than applying a rigid mass production system, production automation is relying more and more on customization of products with highly flexible (mass-) production, which utilizes relatively intelligent machines that can adapt much

more rapidly to product changes and unforeseen process events by maintaining a highly efficient production cycle [6].

I4.0 and relevant initiatives encompass the concept of "edge in the factory," where the decision-making process shifts from cloud to edge. Analytical algorithms can assist in implementing such concepts, as cloud-based solutions are not ideal for complying with the low latency needs of production processes and devices, primarily due to the large volume of data generated at a fast pace [7].

It is expected that knowledge-motivated manufacturing, which becomes a reality by utilizing the big data after processing them with AI analytics methods, will result in the continuing evolution of the industry, and thus in the development of products and services by integrating state-of-the-art communication technologies (e.g., 5G and beyond) into the production. These new technologies are merging physical, digital, and biological worlds. The I4.0 revolution will make people adapt to new living, working, and communicating [8].

An industrial revolution is regarded as the technological progression that enhances the production process. In this sense, the first industrial revolution started exploiting water and steam power in the production process. Then, the second phase of industrialization was reached, starting assembly line usage and electric power in production. Next, a transition to computer and automation systems in the production process was the beginning of the third industrial revolution. Today, the novel integrated production technologies lead to the fourth industrial revolution. They combine the adaptation and usage of several advanced technologies, such as artificial intelligence, nanotechnology, quantum computing, cloud, internet, and robotics. It was triggered by the communication and connectivity based on sensor networks and the internet of things (IoT) [9]. Figure 1 depicts the four stages of the industrial revolution. The terms "digitalization", "computerization", and "second machine age" are also widely used to express these changes in the industry.

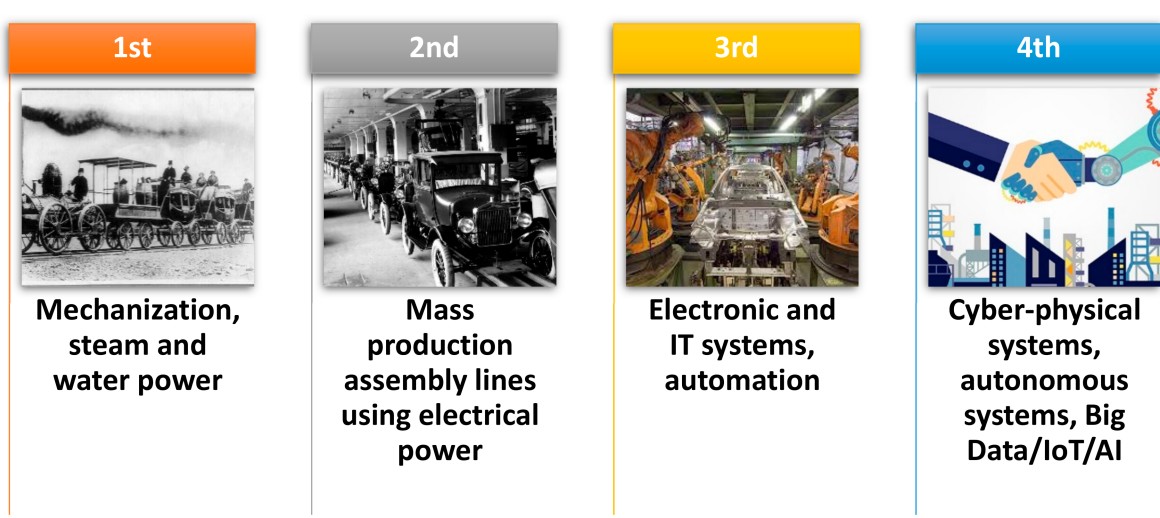

**Figure 1.** The four industrial revolutions.

The adoption of novel technologies under the umbrella of Industry 4.0 and the interconnection of a plethora of devices that become a reality thanks to the integration of 5G networks raise concerns regarding the security and stability against cyber-attacks of these systems. There is not a comprehensive and systematic literature review that addresses this problem. Our goal is to highlight such an issue and attempt to categorize the potential threats related to the technologies that form the Industry 4.0 framework. We could summarize our research question as follows: Which vulnerable elements should be considered in designing and developing an I4.0 network, and how are they connected to the components of I4.0 and the technology provided?

## 2. Main Contribution

This work presents the new technologies that meet the demands and goals of the 4th industrial revolution. However, for the new technologies to verify I4.0's expectations and capabilities, they have to solve several security issues. The integration of a plethora of technologies and elements, despite the provision of multiple interconnected services and the improvement of the production line, forms a complicated interconnected environment where threat management constitutes a challenging mission.

In that sense, we examine the factors that should be considered so the potential mismanagement of threats does not render the untimely end of I4.0 integration. In other words, our goal is to investigate the potential threats that can harm the smooth operation of an I4.0 system, which are the elements that affect, with which security properties are related, and finally, which risks emerge that underlie each I4.0 component.

We present the threats to which each I4.0 element can be vulnerable. Additionally, we categorize the threats based on the goals and the elements of I4.0 that they affect. We present various cyber-attacks that have successfully taken place and prove their impact on the proper function of I4.0. This study aims to be helpful to those designing and implementing systems based on the technologies of I4.0, those examining the flaws of a system, and those trying to shield the backdoors of such a system. Thus, our contribution is crucial, mainly targeting detection and management of I4.0 security risks, security hardening of I4.0 infrastructures (including cyber and physical systems) and risk management methodologies and frameworks.

## 3. Main Aspects of Industry 4.0

The term "Industry 4.0" was raised to highlight the competitive strategy model of European Union industries with other international markets. The model integrates information and communication technology (ICT) and big data analysis into the industry [10]. It is realized through the convergence of the physical and digital worlds. I4.0 is based on the technological developments relevant to the physical, digital, and biological spaces across industries. In this sense, advances in artificial intelligence (AI), nanotechnology, quantum computing, synthetic biology, and robotics will rapidly substitute for the production technologies of the past sixty years. I4.0 revolution will create better prospects in today's production process, including mass customization, flexible production, increased production speed, higher product quality, decreased error rates, optimized efficiency, and better customer proximity [11].

The critical characteristics of I4.0 can be summarized as follows:

- Interoperability: cyber-physical systems (workpiece carriers, assembly stations, and products) allow humans and smart factories to connect and communicate with each other.
- Virtualization: a virtual copy of the smart factory is created by linking sensor data with virtual plant models and simulation models.
- Decentralization: cyber-physical systems make decisions of their own and produce locally (by using 3D).
- Real-time capability: enabling the collection and analysis of data and providing the derived insights immediately.
- Service orientation.
- Modularity: the flexible adaptation of smart factories to changing requirements by replacing or expanding individual modules.
- Convergence.
- Cost reduction and efficiency.
- Mass customization.

In its application and universal understanding of I4.0, the term is most directly related to the world of manufacturing—it could even be called Manufacturing 4.0. This industry is seeing continuous growth and has transformed unlike ever before. In its context to manufacturing, I4.0 is:

- The growth of automation and data technologies powered by the IoT, the cloud, advanced computers, robotics, and people.
- The seamless integration of software, equipment, and people. It extends the speed, reliability, and flow of information between all systems of a manufacturer.
- Key technologies driving I4.0 are:
- autonomous vehicles and robots,
- additive manufacturing,
- distributed ledger systems (such as blockchain),
- big data analytics,
- mobile computing and wearables,
- cloud computing,
- augmented reality.

These technologies affect and enable the creation of different novel business models. I4.0 allows for the creation of smart factories, vertical networks, and horizontal networks.

### 3.1. Industry 4.0 Effect

It is not an overstatement to claim that I4.0 directly impacts other socio-economic fields; it has transformed them towards extensive digitization. The observed fast and remarkable adoption of the Industry 4.0 topic among academic and business entities has led to the general use of the suffix "4.0" to depict Industry 4.0's entry into the systems, processes, and activities. Bongomin et al. [12] conducted a comprehensive review of the neologisms that arose from the original "I4.0" term. Some include Education 4.0, Energy 4.0, Agriculture 4.0, Healthcare 4.0, and Logistics 4.0.

Madsen [13] also examines the plethora of neologisms related to I4.0 through the lens of popularity. He expresses his skepticism against the fact that this concept has gained enormous attention, becoming dominant in public management discourse in such a short period. He claims a trend or a fashion to add the suffix "4.0" to deliver a misleading message for a hypothetical transformation. Such confusion could eventually be an obstacle in the direction of the full integration of I4.0 as it could be "burned out" in concepts that are not compatible with the idea of I4.0 and do not deliver the expected results.

On the other hand, Kovacs [14] seems to be more judgmental against the revolutionary role of I4.0 and the whole discussion about its positive impact on society, especially the expectation of an impressive productivity increase, as he states there do exist inertia forces that ultimately lead to stagnation.

Others claim that the impact of I4.0 is not only restricted to heavy large-scale industries. It functions as the driving force for reforming and leading the small, medium-sized enterprises (SMEs) to the new era. The constraints in resources render SMEs less flexible in adopting and following technological advancements. SMEs face issues in technology implementations concerning data management, data extraction, and the organization's functional structure. Various I4.0 factors have a positive role in promoting sustainable business performance in SMEs [15].

Many scholars now study the effect that the adoption of I4.0 may have regarding environmental and, more generally, sustainability-related issues. For instance, Ferrari et al. [16] propose an architecture that provides a valuable tool for evaluating and monitoring environmental impacts related to the production process in a fully digitized factory according to Industry 4.0 criteria. Gupta et al. [17] developed a framework for the evaluation of sustainability performance. Their findings indicate that adopting practices of the circular economy, cleaner production, and Industry 4.0 by manufacturing organizations improve sustainability performance.

Sony [18] conducted a thorough systematic literature review to collect and summarize the reported advantages and disadvantages regarding the I4.0 concept. Table 1 mentions the main pros and cons, while the reader can refer to the related article for an extensive analysis.

Table 1. The main pros and cons of I4.0.

| Advantages | Disadvantages |
|---|---|
| • Strategic competitive advantage | • The negative impact of data sharing in a competitive environment |
| • Organizational efficiency and effectiveness | • Total implementation of Industry 4.0 is necessary for the success |
| • Organization agility | • Handling employees and trade unions apprehensions |
| • Manufacturing innovation | • Need for highly skilled labor |
| • Profitability | • Socio-technical implications of Industry 4.0 |
| • Improved product safety and quality | • Cybersecurity |
| • Delightful customer experience | • High initial cost |
| • Improved operations | |
| • Environmental and social benefits | |

I4.0 represents a dynamic field with a direct impact on various domains, extended to affect the socio-economic life; consequently, any controversy is significant to emerge.

3.1.1. Evolution of I4.0 by Adapting Edge Technology

The manufacturing industry is always driven by the evolution of technology—from steam engines to electricity, microprocessors, computers, automation, and recently, artificial intelligence, IoT, and cyber-physical systems. Several initiatives have been launched globally over the past few years (e.g., Industry 4.0 in Europe [19]; Made in China 2025 [20]; U.S. Advanced Manufacturing [21]) to address the global needs for higher efficiency, lower costs, and mass-personalization of products and services. The synergetic effect between emerging technologies and needs has led to the creation of new manufacturing paradigms characterized by [22]:

i. digitalization and integration of manufacturing resources on cloud-based platforms as adaptive, secure, and on-demand services,

ii. intelligent and connected objects capable of real-time and autonomous decision-making via embedded electronics and analytical/cognitive capabilities.

Examples of recent paradigms include smart manufacturing [23], cyber-physical production systems (CPPS) [24], I4.0 [19], and cloud manufacturing [25]. We henceforth refer to these new paradigms as smart manufacturing [26].

I4.0 is related to the "smart factory". In a smart factory, physical systems can cooperate and communicate with each other and with humans in real-time, and all of them are enabled by the IoT and related services that allow a production factory to manage itself virtually [27]. Zheng et al. [28] define smart manufacturing systems as "fully integrated and collaborative manufacturing systems that respond in real-time to meet the changing demands and conditions in factories and supply networks and satisfy varying customer needs". The primary sources of data gathering and analysis in I4.0 are the intelligence of industrial applications that constitute innovative design, smart machining, efficient monitoring, smart control, and intelligent scheduling [28]. The vision of I4.0 is clarified through the set of three pivotal examples [29]:

i. *The intelligent product.* The products themselves can order production resources and coordinate the manufacturing process for its accomplishment.

ii. *The smart machine.* The machines become cyber-physical production systems. Decentralized, adaptable, jointed, and self-regulating production networks substitute conventional production structures.

iii. *The augmented operator.* I4.0 project's goal is not to create production plants without workers. It instead aims to acknowledge the crucial role of the human factor: taking advantage of technology, the human operator is the most adjustable part of the production system.

In this direction of rendering the smart product an active member of the production process, the latter should convey its existence, qualities, and needs to the ambient machines or humans. Here comes the IoT that functions as the facilitator for providing the means for these requirements. Subsequently, IoT relies on cloud computing, its variations, and the omnipresent digital communication infrastructure for effective and economically viable global information transfer [30]. Moreover, another critical point related to the smart products in I4.0 is that they constitute a continuous source of data regarding themselves, the environment in which they belong, and the interaction with the user [31]. Big data infrastructures and human–computer interaction are the domains on which the attributes mentioned above rely to find their technological enablers. Furthermore, big data infrastructures and human–computer interaction, in turn, rest on cloud computing and artificial intelligence.

The smart machine can self-organize to satisfy the production needs obtained from the smart product and the production environment [29]. A significant element of this dynamic production network concerns the generation, storage, and distribution of energy [32]. The smart machine's principal technological enabler originates from robotics, and here again, IoT provides the essential communication means. Furthermore, AI makes available the means for advanced reasoning and independent behavior. The blockchain embodies the vision of the future machine-to-machine communication and autonomous behavior, providing a seamless record of incidents and automatic transaction implementation without trusted entities. Blockchain technologies utilize the global digital communication infrastructure for effective long-distance communications.

Considering all those mentioned above, it is evident that the human presence is still vital, albeit it needs to be reconsidered, exploiting the tools that render the operators become augmented operators. We cannot ignore the interaction between the operators and the working environment. Advancements in human-computer interaction and robotics allow us to design collaborative working environments where humans may interact using virtual/augmented reality technologies with enabled cyber-physical systems (CPS) empowered by AI [30].

Cyber-physical systems and the IoT enable smart factories. Cyber-physical systems create a virtual factory copy through sensors and actuators [4], enabling decentralized decision-making. Interconnections allow for collaboration between machines, between machines and people, and between people. The smart factory has an element of "consciousness" through artificial intelligence that enables it to make decisions about its own maintenance and manufacturing processes (self-optimized). Smart factories will expand workers' roles, moving beyond routine work that will require decision-making and a more comprehensive range of skills, thereby changing the way people work.

Smart manufacturing is a broad manufacturing category that employs computer-integrated manufacturing, high levels of adaptability and rapid design changes, digital information technology, and more flexible technical workforce training [33]. Other goals sometimes include fast changes in production levels based on demand, optimization of the supply chain, efficient production, and recyclability. A smart factory has interoperable systems, multi-scale dynamic modeling and simulation, intelligent automation, strong cybersecurity, and networked sensors in this concept.

The broad definition of smart manufacturing covers many different technologies. Some of the critical technologies in the smart manufacturing movement include big data processing capabilities, industrial connectivity devices and services, and advanced robotics.

The evolution of products into intelligent connected devices is radically reshaping companies and competition. All smart connected products have three core elements: physical, smart, and connectivity components [31]. Key features of the smart factory are connectivity, optimization, transparency, proactivity, and agility [34].

### 3.1.2. Evolution of I4.0 by Adapting Mechanical Advances

Mechatronics is a crucial factor in manufacturing. Mechatronics is "the synergistic integration of mechanical engineering with electronics and electrical systems using intelligent

computer control to design and manufacture industrial products, processes, and operations" [35] to produce sturdy, low-cost, more trustworthy, more adaptable, and versatile goods. Mechatronics systems have met significant development since their introduction by the Japanese engineer T. Mori (Yaskawa Electric Corporation, Yahatanishi-ku, Kitakyushu, Japan) in 1969. The prefix "Mecha" stands for mechanical, and the suffix "tronics" stands for electronics, which embodies information technology (IT).

The mechanical advances (e.g., machine tools and power generators) intend to diminish or ideally eliminate the human's physical stress. On the other hand, electronics technology and IT intend to limit human mental strain [36]. In this sense, mechatronics can relieve humans from physical and mental strain in various aspects of life. The logic behind mechatronics is to incorporate various technologies from conventional interdisciplinary domains in engineering and computer science, aiming to develop well-established stand-alone systems in new products. New opportunities for mechanical design and automatic functions arise after mechanical systems incorporate microelectronics and information technology [37]. The turning point that accelerated automation in the industrial sector was the invention of PLCs (programmable logic controllers) by General Motors in 1968, along with the other controllers, such as PID (proportional-integral-derivative) and SCADA (supervisory control and data acquisition) [38].

## 4. Guarding Industry 4.0 Components

Several intrinsic security issues emerge from the nature of I4.0 itself, and the technologies incorporated are the ones that render the I4.0 a promising framework. When dealing with cutting-edge technologies in the production line, it is known that the benefits of utilizing the new capabilities of interoperability create a fertile ground for cyberattacks. Therefore, in the direction of the total and safe integration of I4.0, security issues should seriously be considered.

Here, we present our work concerning the potential of the 4th industrial revolution through the angle view of essential enabling technologies. After conducting the literature review, we demonstrate the results and necessary directions [39]. We focus in light on the security issues that come as a result of enabling these technologies. We could summarize our research question as follows: Which vulnerable elements should be considered in designing and developing an I4.0 network, and how are they connected to the components of I4.0 and the technology provided?

The main components of I4.0 are illustrated in Figure 2. Each of them is analyzed consistently, focusing on the security issues that arise from adopting each technology.

### 4.1. Cyber-Physical Systems (CPS)

The idea of CPS arose due to the convergence among IT, embedded computing and mechatronics. The first one is indicated by the frequent use of terms such as cyber, computation, networks, and software. The second one is reflected in terms such as physical, integration, components, control, processes, engineering, and systems. This convergence of domains has led to research challenges and associated innovation possibilities related to the concepts of hybrid physical and logical systems [40].

In the concept of CPSs, physical objects and systems are amalgamated with integrated computing facilities and data storage. CPSs can create networks in which data and information are shared and exchanged with other objects and systems. Briefly, CPS unites two worlds, adopting and embracing their developments. From one side, we have the plethora of machines, facilities, and networks that emerged as the outcome of the Industrial Revolution. On the other side, we witness the recent progress in computing, information, and communication systems that Internet Revolution actualizes [41].

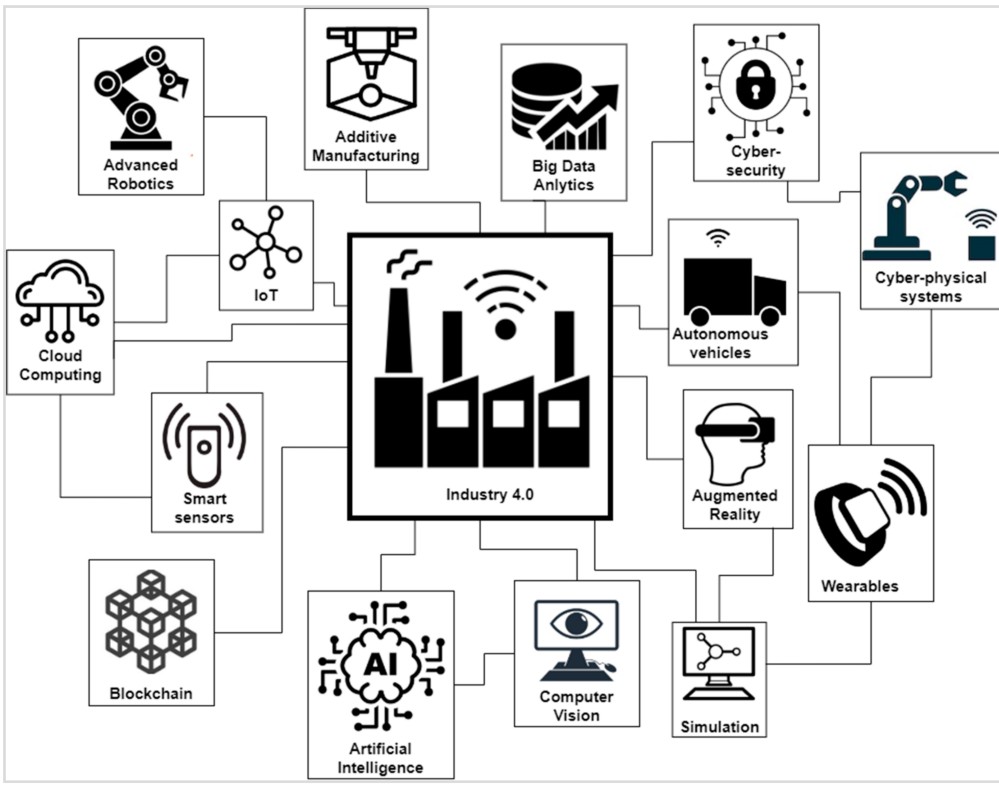

**Figure 2.** Key components of Industry 4.0.

All the features of a smart factory with sensors, network, and legacy systems are mirrored in the tablet or PC, integrated with big data analytics in the cloud or central server. All the actions and tasks in the production and manufacturing cell are monitored transparently in real-time. Based on big data analytics, quality issues or facility problems are forecasted or prevented proactively. The results are fed back to the industrial field to monitor and control the process [9]. All the elements of such a fully integrated network should be *smart objects* that can communicate and interact and eventually achieve a common objective [42].

**Threats in Cyber-physical systems:** IoT-enabled CPS are facing vulnerabilities and threats from the internet. Unfortunately, this physical process interaction for providing real-time sensing, dynamic control, and information services for complex systems opens, in turn, the door to multiple kinds of attacks in the supply chain and leads to diverse security problems; many of them related to confidentiality (mainly in the theft of intellectual property), integrity, and availability of the product life cycle [43].

### 4.2. Internet of Things (IoT)

The term "Internet of Things" has its roots in a presentation of Kevin Ashton in 1999 to Proctor and Gamble and the relevant work in the MIT Auto-ID Center [44]. The whole idea of IoT came out from the RFID (radio-frequency identification) field, and at first, it emphasized the ability to track the location and status of any physical object, especially in supply-chain applications. The IoT is a concept closely related to ubiquitous computing, and due to its direct connection with the networking and information technology communities, commonly used terms include information, communication, networks, connectivity, and data [40].

The internet has a pivotal role within the concept of I4.0. Therefore, it is bequeathed with the problems and issues concerning the internet as an entity, such as security, privacy, availability, infrastructural costs, performance, network neutrality, the evolution of the protocols, fault detection, and attacks. In other words, we can regard the internet as the

most fundamental enabler of I4.0. This is apparent because we cannot visualize the concept of I4.0 without the internet as we know it [30]. It plays the role of bringing together distributed entities (humans and machines). The intelligence comes from understanding the surrounding environment, involving information exchange using the communication infrastructure, or by taking advantage of the computational capacity and memory storage in distant cost-effective datacenters (i.e., the cloud), which are accessed via the internet [30].

The IoT is defined as the interaction between technologies, including smart objects, machine-to-machine communication, radio-frequency technologies, and a central hub of information to monitor physical objects' status, capture meaningful data, and communicate that information through IP networks to software applications. In the IoT, objects are equipped with auto-ID technology to detect and monitor and collect required data. Utilizing this technology enables the users to collect and analyze any data, such as temperature, changes in quantity, or other types of information through wireless communication and make efficient and accurate decisions [45].

Integrating digital capabilities, for instance, network connectivity and computational capability, with physical devices within the contexts of CPS and IoT provides performance and functionality improvement. Many systems embody this integration, e.g., from smart vehicles and smart grids to advanced manufacturing and wearable medical devices. Overall technological progress and economic growth occur in diverse fields, such as energy and transportation to health care, agriculture, public safety, and smart cities. [40].

The current CPS and, more generally, IoT systems suffer from the inherent problem of centralized architectures. In this direction, efforts are made to expand these centralized architecture models and ultimately comply with the needs of future CPSs. The remedy for such problems is the decentralization provided by consensus-driven distributed ledger technology, such as the blockchain, combined with cryptographic processes. The combination of CPSs/IoT and blockchains will provoke a disorder to processes across various fields, such as manufacturing, agriculture, banking, transportation, shipping, energy, and health care. Nevertheless, there is yet a lot to be done focusing mainly on privacy, scalability, performance, and potential economic advantages [41].

### 4.3. Industrial IoT (IIoT)

The European Telecommunications Standards Institute (ETSI) defines IIoT as machine-to-machine or machine-type communication [41]. It is integrating IoT into the concept of I4.0, which is described as "machines, computers, and people enabling intelligent industrial operations, using advanced data analytics for transformational business outcomes" [46]. In broad terms, it is explained as industrial machines that bear sensors connected with other machines via the internet for monitoring, analysis, and management.

**Threats in IoT:** Most industrial IoT devices are usually built from many components manufactured by multiple suppliers without a common standardization security requirement. This variety of different IoT elements raises several security and privacy issues. How will an industrial partner or an industrial company trust the hardware devices or their embedded software? How can a company trust the installed software (see the Ericson faulty software event [47])? If a faulty sensor advertently/inadvertently gives wrong information about the state of the process, the industry will identify the fault or claim reimbursement.

Moreover, the industrial IoT network interconnects various heterogeneous components for providing online information processing. A malicious user could exploit this real-time interaction and execute his cyberattack on a machine (such as a vehicle, robot) or an IoT device via the internet. The IoT framework suffers from several security issues, which are more challenging than any other case because of the complex and resource-constrained IoT device environment. There are many research efforts ongoing to investigate and provide efficient security solutions for the IoT environment [48], particularly to address resource constraints and scalability issues [49]. A comprehensive top-down survey of the most recent proposed security and privacy solutions in IoT is provided in [50].

## 4.4. Big Data Analytics

"Big data" is a term with a broad meaning and definition. Its initial meaning referred to datasets characteristics regarding the existing technologies. Over time its meaning shifted, and now it represents the technologies designed to extract economic value from massive volumes of a wide variety of data by enabling the high-velocity capture, discovery, and/or analysis [51]. The well-known five V's representation offers a comprehensive summary of big data's characteristics:

i.　　　Volume (refers to the unimaginable amounts of information generated every second).
ii.　　Velocity (refers to the speed at which the data is generated, collected, and analyzed).
iii.　　Variety (entails the processing of diverse data types collected from varied data sources).
iv.　　Veracity (means the degree of reliability that the data has to offer).
v.　　　Value (refers to the ability to transform an enormous amount of data into something that brings profit).

Specifically, if we focus on the product lifecycle, it can be determined that all its phases provide their own categories of input data to be processed with big data tools and produce output data for decision making [52].

**Threats in big data:** Big data management is crucial for managing the production line by performing computations on large volumes of data. The access control to these large data volumes should follow strict access control policies and rules that apply to all components, CAD, and IoT. Moreover, misapplication of a global access control mechanism may lead to fraud or privacy issues. A corrupted employee could replica some crucial information, or grand access users may access the employee's personal information.

## 4.5. Artificial Intelligence (AI)

Artificial intelligence refers to the simulation of human intelligence in machines programmed to think similarly to humans and mimic their actions. The term is applied to any machine that exhibits traits associated with a human mind, such as learning (i.e., acquiring information and rules for using it), reasoning (i.e., adopting rules to reach conclusions), and self-correction.

The field of AI was formally founded in 1956, at a conference at Dartmouth College, in Hanover, New Hampshire, where the term "artificial intelligence" was coined. Nowadays, AI is a general term that embraces diverse, interconnected domains, such as robotics, big data analytics, machine learning, machine/computer vision, and natural language processing [53]. Machine vision is related to capturing and analyzing visual information utilizing cameras, analog-to-digital conversion, and digital signal processing. The ultimate goal is to develop a model that represents the natural world from images [30].

Machine learning is an application of AI that provides systems the ability to learn and improve from experience without being explicitly programmed automatically. Machine learning focuses on developing computer programs that can access data and use it to learn for themselves. It includes three types of algorithms:

- Supervised,
- Unsupervised,
- Reinforcement learning.

Deep learning is a subfield of machine learning concerned with algorithms inspired by the structure and function of the brain called artificial neural networks. Multilayer ("deep") architectures are the critical aspect of deep learning. Pattern recognition is also a branch of machine learning that finds regularities and similarities in using machine learning data.

**Threats in AI:** Artificial intelligence introduces new autonomous and learning processes in the industrial sector and raises many security and privacy issues. Malicious users could modify learning models or pre-processed distributed data to gain the desired output or extract critical/privacy information from the deployed AI models.

*4.6. Advanced Robotics*

Today, robots play a crucial role in human life, from industrial manufacturing to healthcare and transportation [54]. The distinctive property to classify a system in the field of robotics is whether it can perform the three following functions characterizing a robot:

i.　　acting on environmental stimuli,
ii.　　sensing,
iii.　　logical reasoning.

It is not an exaggeration to state that robots, as they can automatically execute complex series of actions, have been considered one of the third industrial revolutions' cornerstones. Today, robotics met with the significant progress that led to evolutionary robotics, which applies the selection, variation, and heredity principles of natural evolution to the design of robots with embodied intelligence. It represents the automatic creation of autonomous robots that exploits the tools from the fields of neural networks, genetic algorithms, and dynamic systems [55].

*4.7. Cloud Computing*

Cloud computing (an enabler of utility computing) delivers different services through the internet in real-time, with minimal interaction with the provider [30]. These resources include tools and applications, such as data storage, servers, databases, networking, and software. The concept of cloud computing is implied in many I4.0 enablers, particularly in big data, IoT, and visual computing. Furthermore, "cloud manufacturing", is conceived as a "networked manufacturing model based on on-demand access to a shared collection of distributed manufacturing resources (instead of just computing/storage as in classic cloud computing)" [56]. The objective is to develop production lines that are transient, adaptable, and distributed, which can allocate resources to customer requests optimally. The lifecycle costs and the total delays are minimized by this process, whereas a user-customized product is supplied—cloud manufacturing shares common goals with I4.0 [56].

Even though cloud manufacturing incorporates the concept of cloud computing, humans are not only not excluded from the first one but also are vital participants [57], in contrast to the latter, where humans are kept out. From this perspective, cloud computing is just one convenient technology enabling the service-oriented architecture based on the cloud manufacturing paradigm [58].

**Threats in the cloud environment:** The cloud provides various valuable services in I4.0 services, but cloud architecture is vulnerable to various attacks, such as information and service theft (e.g., through virtualization vulnerabilities) and infrastructure availability.

*4.8. Fog-Edge Computing and Mobile Cloud*

Fog computing is a layered model for enabling ubiquitous access to a shared continuum of scalable computing resources. The model facilitates the deployment of distributed, latency-aware applications and services and consists of fog nodes (physical or virtual) residing between smart end-devices and centralized cloud services [9,59]. It is considered an extension of cloud computing to the edge network, providing services closer to near user's devices instead of sending data to the cloud. Edge computing also allows computation to be performed at the edge of the network but closer to the data sources [60]. Edge computing is a decentralized computing infrastructure where computing resources and application services are distributed along the network from the sensor node to the cloud [9].

*4.9. Virtual and Augmented Reality (VR/AR)*

In I4.0, the operation, reporting, and monitoring of data via integrated data chains could be visualized in the virtual reality environment. The latter can integrate industrial environments, redesign, retest, and refine them for realizing the Virtual industrial simulation. VR has to deal with specific challenges and issues to gain the visualization part of

I4.0 [61,62] by providing new solutions and more effective possibilities for an innovative working environment.

Industrial companies must manage various technology factors for reshaping the industrial environment in order to adapt VR/AR technology:

- The cost of adopting VR/AR technology is not negligible and should be taken into account. Several technologies exist to get access to the virtual world, e.g., Oculus Rift, HTC Vive. The cost of each tool depends on the extent of immersion it provides. The better the immersion, the more expensive it is. The virtual environment is servers providing VR/AR tools, which are accessed from everywhere using the client-server approach. The integration of VR/AR tools in I4.0 demands assessing the cost factors and the benefits of the specific VR/AR technologies and servers.
- A matter of investigation and consideration is finding how to use simulation and VR/AR models to produce reliable approaches for adapting physical engines in I4.0. Physical movements not close to reality may negatively impact or give wrong results when simulations refer to user interaction with the environment.
- The nature of VR/AR realization in the industrial environment is essential. Users need time to adapt to the VR/AR environment if they use glasses; otherwise, they will feel lost and sick if they use them. The VR/AR methodology (3D glasses and VR booth) should be user-friendly.
- VR/AR is used for real-time visualization of specific virtual models and simulation results, but it has to be considered the adaptation factor of known and implemented virtual models to new events and circumstances. This requires a viable use case scenario and serious programming effort. In order to maintain the VR/AR scenarios existing code needs updating, which is time-consuming and challenging for users with little to no coding experience. Consequently, design adaptable virtual models need a lot of time and effort.
- Furthermore, we have to consider connectivity issues of the visualization software with the connecting parts (physical systems, embedded systems, sensors, actuators, electronic hardware, software, etc.) Sometimes, VR/AR online software may lag if the user has a poor connection, resulting in poor performance and poor experience. The software needs to be optimized to reduce the stress in both the internet connection and the computer's graphical S&W to provide a smoother experience.
- Ensuring that VR/AR models will produce real-time concrete results. The VR modes must deal with complex real-time events and reproduce the VR representation based on simulation feedback.

**Threats in VR/AR environments:** VR/AR application in I4.0 allows companies to decrease design and production costs, maintain product quality, and reduce the time needed to go from product concept to production. Unfortunately, VR/AR may not realistically simulate a real-life event, and a malicious user could use this fact for succeeding incorrect, invalid decisions, and inappropriate actions

### 4.10. Digital Twin

Digital twin is the virtual representation of a physical product that contains information about this product. The origins of that term lie in the field of product lifecycle management [63]. Grieves [63] describes the digital twin as being composed of three elements:

- a physical product,
- a virtual representation of the product,
- the bi-directional data connections, from the physical to the virtual representation, and vice versa.

The digital twin can tackle the challenge of seamless integration between IoT and data analytics by creating a connected physical and virtual twin [64]. A digital twin environment allows for rapid analysis and real-time decisions made through accurate analytics.

Grieves depicted this flow as a cycle between the physical and virtual world (mirroring or twinning). Figure 3 depicts the cycle where the data stream is directed from the physical to the virtual space. On the other hand, the information and processes are directed from the virtual to the physical space [65].

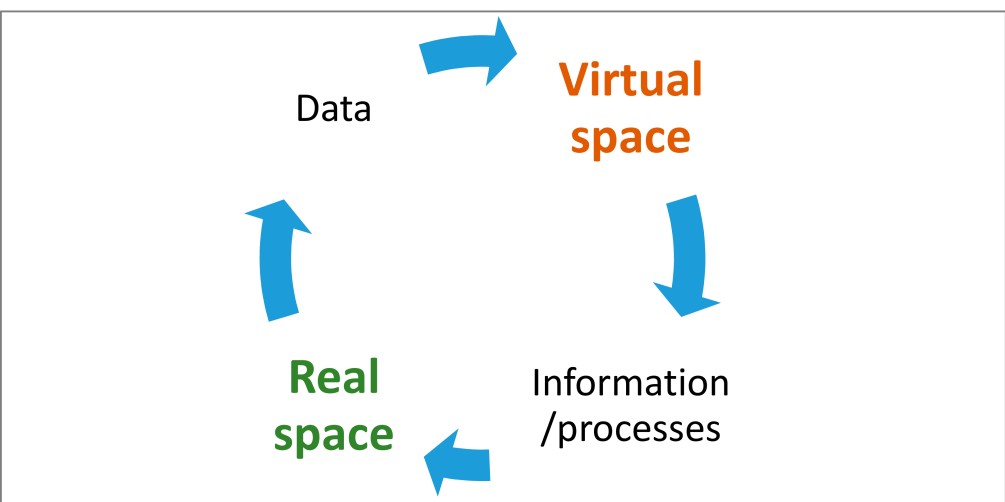

**Figure 3.** Key components of digital twin.

The following value additions characterize digital twin [66]:

1. Real-time remote monitoring and control.
2. Greater efficiency and safety.
3. Predictive maintenance and scheduling.
4. Scenario and risk assessment.
5. Better intra- and inter-team synergy and collaborations.
6. More efficient and informed decision support system.
7. Personalization of products and services.
8. Better documentation and communication.

**Threats in digital twin:** Product optimization is introduced by enabling a digital model of a physical object that simulates the object in a live setting. Unfortunately, the digital twin expands the attacker's capabilities since it can control the physical world through bidirectional interfaces. If a hacker obtains the twin, it can guide the system, identifying components, behaviors, and interfaces. This immediately gives the hacker an internal view of the system to be attacked and will help him identify vulnerable attack points. Once a hacker compromises a digital twin, he can easily spoof the twin's behavior or even access the physical system.

*4.11. Blockchain*

Blockchain is a promising technology in the information and communications domain. Blockchain technology has been seen as one of the most important innovations since the internet and this century. The blockchain is a decentralized digital database (ledger), which stores the transactions committed by users. The connected community (miners) verify the authenticity of such transactions before adding them to the ledger. Thus, the blockchain employs a distributed trust model by eliminating a third-party centralized trust model [41].

*4.12. Horizontal and Vertical System Integration*

Integration renders a central idea in I4.0, and it is seen in three different axes:

i. horizontal integration that is the collaboration between enterprises along a value chain;
ii. vertical integration that is the extensive automation inside an enterprise,

iii.　end-to-end integration envisions connections across the value chains between every couple of digitally enabled participants (machine-to-machine, human-to-machine, human-to-human) [67].

*4.13. Cybersecurity*

Cybersecurity in CPS/IoT systems is different from that in conventional ICT-only systems in three main categories:

- complex cybersecurity deployment landscapes,
- cyber-attacks on physical systems,
- physical attacks on and physics-based cyber-system mitigation.

Firstly, the amount and variety of entities used in CPS/IoT systems present considerable challenges [68]. The addition of networked connectivity to everyday objects increases the number of points of attack that must be protected. Secondly, integrating logical and physical components means that an attack on IT systems can gain control over critical physical systems. Cybersecurity in the era of I4.0 directly impacts the CPSs, internet of services, and IoT [69].

## 5. Security Challenges in Industry 4.0

The goal of I4.0 is to overlay a simulation on a real-time production line that will bring countless new desirable benefits to market sectors, but this new industrial path will not be adopted if it will not address and solve the potential security and privacy issues. Security is a significant concern for novel interconnected industrial environments since industrial systems are not designed to be connected with the outside world and deal with the internet, which is not secure. I4.0 faces a significant barrier related to cybersecurity threats because this new industrial ecosystem could expand the attack space in the entire ecosystem, not only at a physical level but also at a logical level. The importance of "digital security" has been reported widely [47]. Safeguarding data, especially when heterogeneous technologies are merged, is I4.0's key concern [70]. The industrial market is already aware of security problems in the current industrial environment since companies, such as Trend Micro [22] or Kaspersky Lab [71], have officially reported the lack of security in the operational industrial interconnected environments.

Here, we map the security and privacy challenges and threats for ensuring security in I4.0 manufacturing and identify the most critical assets in this regard. Initially, the threats faced by the I4.0 ecosystem are classified. Various actors providing innovative services are involved in the new industrial ecosystems, so this environment is becoming too attractive for various malicious actions that offer different incentives. It is essential to address the threats involved in the industrial ecosystem because one must identify how much trust one can afford to place and what residual consequences one can accept to define appropriate security levels.

*5.1. The Main Threats in Industry 4.0*

I4.0 includes various technologies to deliver innovative methodologies, advanced services, and new possibilities in the industrial market. Below, we identify the main categories of threats for each specific technology:

**Deception threats:** Industrial competitors could intentionally organize or support someone, preventing an industrial physical part from operating normally or a logical part to deliver a service. Various physical and digital entities perform actions to conclude a service in the I4.0 ecosystem. Thus, a malicious insider action could cause service disruption. These threats are crucial because they cause substantial financial and reputation loss but are pretty standard since corrupted personnel or industrial competitors will violate the regular operation of a critical component for gaining economic profits.

**Threats to industrial products:** The vision of I4.0 is to build an industrial ecosystem capable of managing and controlling the production chain. If someone gains unauthorized access and controls the product construction, he can easily harm the product's construc-

tion line by stealing products, altering crucial components, or changing sipping/labeling products' information.

**Cyberattack threats:** The heterogeneous architecture of the I4.0 ecosystem is based on incorporating various IT components and IoT devices. Every IT component must address various cyberattacks, such as malware infection, unauthorized data modification, and malicious updates, to prevent critical data and information disclosure. Exposure of critical data is essential, such as revealing the design of a new product, which could cause substantial financial and reputation loss.

In Section 4, we have presented various security issues that emerge from the nature of I4.0 itself and the technologies incorporated. This section shows how various threats can lead to economic losses for the associated companies and industries and how they can affect several economic and social aspects of the future industrial paradigm's. Our focus is to point out a need for further research to make the future Industry 4.0 ecosystem accomplish its promising benefits in the economy and society.

To evaluate the security risks on each component of I4.0, we use the following known security properties:

- Confidentiality: All the produced physical or logical data from all the I4.0 components must not be accessed by unauthorized entities. If the confidentiality property is not preserved, the produced products or data may be stolen.
- Integrity: All the provided services from all components of I4.0 must preserve their integrity. When someone compromises the integrity property, this compromise may stay hidden while being exploited constantly.
- Availability: All the provided services from all the components of I4.0 must be available by authorized users and services. If the availability property is not preserved to all I4.0 components, the produced products may be lost, and services, data, and products may be unrecoverable.

*5.2. The Security Threats for Main Issues of Industry 4.0*

Below, we identify various threats that can be applied in the production line of I4.0 for affecting the standard functionality of each component:

- *Threats in Operation of I4.0:*
  - *Failure to run a requested task*: Attacks can prevent an active function from being finished in the production line, and the participating companies/parties are always vulnerable.
  - *Failure or holding back to run planned tasks*: If one of the participating members/companies of I4.0 is exposed to continuous attacks, it is infeasible to recover.
  - *Financial and reputation loss and lack of consistency*: In I4.0, the participating entities are committed to terms and legal conditions that may not be satisfied.
  - *Loss of trustworthiness*: Market relations of the participating entities are fundamental for utilizing the production line of I4.0, and these relations are based on trust between entities/companies. This established trustworthiness may not be preserved if an entity could not accomplish a task without uncontrolled or unmanaged threats.
- *Threats in Components of I4.0:*
  - *Malfunction of the infrastructure of I4.0*: Malicious users could take advantage of I4.0 vulnerabilities to damage the physical facilities of I4.0.
  - *Malfunction in information systems or networks*: Cyber-attacks, such as ransomware, could relentlessly disable the underlying IT infrastructure that supports the I4.0 environment
  - *Malfunction in parts*: Managing threats to a hugely dynamic, interconnected environment of I4.0 becomes trying to manage the threats against physical/digital assets due to the supply chain transforming into a chaotic supply web.

- - *Stealing or modifying the produced data*: The data produced and transmitted on the dynamic environment of I4.0 can be tampered with by malicious adversaries; this attack could lead to loss of companies' intellectual property and makes it difficult to capitalize companies' know-how.
    - *Loss of trustworthiness*: This highly dynamic environment helps any participant company to steal the intellectual property of any other participated entity.
  - *Threats in participating entities:*
    - *Threatening user's safety*: If customers could succeed in modifying the production methodology of I4.0, these may cause accidents and threaten the user's safety.
    - *Decreasing people's trust*: Customers are the final point of I.0.4, and they have to trust the products of I4.0. Thus, the trustworthiness will be decreased if the customers know that someone could interfere with the production line and not trust the quality/safety standards of the produced goods.
  - *Threats to economic/social relations:*

  The high dynamic, interconnected environment of I4.0 could affect the relations of the participating entities if it becomes an untrusted environment.

  Table 2 summarizes the main reported threats to specific sections of I4.0. It indicates for each threat incident which security property is affected and in which category it belongs.

**Table 2.** Categorization of Reported Threats Events in I4.0.

| Security Property | | | Component of I.0.4 | Threat Category | Threat Event | Reference |
|---|---|---|---|---|---|---|
| *Confidentiality* | *Integrity* | *Availability* | - | - | - | - |
| - | x | - | CPS, IoT, Cloud | Threats in operation of I.4.0 | Countries use cyber threats for controlling production lines | [72–75] |
| - | x | x | CPS, IoT, Cloud | Threats in operation of I.4.0 | NotPetya attack | [72,76] |
| - | x | x | CPS, IoT, Cloud, VR/AR | Threats in operation of I.4.0 | Energy sector Attack | [77] |
| x | - | - | CPS, IoT, Cloud | Threats in components of I.4.0 | Hacking service providers | [78] |
| x | x | - | CPS, IoT, Cloud | Threats in participating entities | SW hijacked for installing backdoors | [79–81] |
| x | x | - | CPS, IoT, Cloud | Threats in participating entities | Online credit card skimming attack | [80,82,83] |
| x | - | - | IoT | Threats in participating entities | Harvesting data via SDK | [84] |
| x | - | - | CPS | Threats in components of I.4.0 | Compromise S/W for harvesting Data | [85] |
| x | - | - | CPS | Threats in components of I.4.0 | Compromise S/W in industrial control systems | [85] |
| - | x | x | CPS | Threats in operation of I.4.0 | Exploiting insecure SCADA systems | [86] |
| x | - | - | - | Threats in participating entities | Compromised employees | [87] |
| - | x | - | CPS, IoT, Cloud | Threats in components of I.4.0 | Antwerp port smuggling | [72] |
| - | | x | CPS | Threats in components of I.4.0 | Eli Lilly warehouse theft | [73] |
| - | x | - | CPS, IoT | Threats in participating entities | Contamination of meat products | [74] |
| - | x | x | - | Threats in participating entities | Explosive printer cartridges | [75] |

There are many incidents where countries' secret services manipulated the production line via cyber threats [72–75]. The National Security Agency (NSA) was accused of compromising encryption mechanisms belonging to security vendors and inserting backdoors into final products. China tried to modify Lenovo motherboards' production line to infect them with malicious software [73]. Russia has also been accused of injecting malware via a legitimate SW component [74,75]. Moreover, Chinese spies tried to steal private information by hacking eight of the world's biggest technology service providers [78].

The new wave of cyberattacks attacked the energy sector in Europe and North America, and they tried to disrupt affected operations severely. The group behind these attacks is known as Dragonfly. The attackers try to insert a Trojan and disrupt operational ac-

tions [77]. According to Symantec, the attack was based on compromised websites and hijacked software updates.

The NoPetya cyber-attack took down thousands of computers in dozens of countries, disrupting shipping and businesses by exploiting a vulnerability of a typical software component used by most Ukrainian companies. The malware rapidly extended to take down thousands of computers in dozens of countries, disrupting shipping and business continuity, causing a total economic loss of more than10 billion [72,76]. In 2017, the Kaspersky Lab discovered a backdoor in a server management SW product used by hundreds of large businesses worldwide, among them: energy firms and pharmaceutical manufacturers [79]. The malicious code, known as ShadowPad (belonging to NetSarang), allowed attackers to download and execute malicious modules and/or steal sensitive data remotely. The ShadowHammer cyber-attack was launched in 2019 to integrate malicious code into ASUS Live Update Utility components. Through this infection, attackers could install backdoors on millions of devices and connect them to predefined targets [80,81]. In 2019, a skimming code was injected into the shared JavaScript libraries of the e-commerce PrismWeb platform, impacting 201 online university campus stores in the US and Canada [72,73]. That same year, Trend Micro also detected and blocked another malicious skimming code loaded on 277 e-commerce websites. Originally and according to the investigations, this latter code was not planned to be injected into e-commerce websites directly. However, instead, it was designed to infect the JavaScript library belonging to a French online advertising company called Adverline (serving as a "third-party" entity). In both incidents, the malicious codes aimed to steal payment data and later send this information to one or several remote servers. On April 14, 2019 the attackers injected a script into the payment checkout libraries used by the PrismWeb platform. The notorious online credit card skimming attack is known as Magecart. The attack, facilitated by a new cybercrime group, impacted 201 online campus stores in the United States and Canada [80,82,83].

In 2017 hackers broke into British company Piriform Ltd.'s free software that optimizes computer performance, potentially allowing them to control the devices of millions of users, the company, and independent researchers [85]. This attack targeted recollecting information and gaining access, capabilities, and resources, leading to new attack vectors. Check Point Research has discovered a group of Android applications massively harvesting contact information on mobile phones without the user's consent. The data-stealing logic hides inside a data analytics software development kit (SDK) seen in up to 12 different mobile applications and has so far been downloaded over 111 million times [84]. In 2014, the organized Russian group Black Ghost Knifefish compromised software produced by three industrial control systems equipment providers in Germany, Switzerland, and Belgium. The goal was basically to extend the malware and compromise countless other victims throughout Europe [85]. In 2018 exploiting vulnerabilities of third-party systems in supervisory control and data acquisition (SCADA) environments lead to a cyberattack that succeeded in shutting down numerous pipeline communication networks [86]. Last but not least, it is very often to compromise an employee in I.0.4 since he may live in a country with low living standards [87]. In 2016 three Wipro employees in Kolkata have been arrested in connection with a security breach in the customer records of a UK-based telecom client TalkTalk, a development that had major implications for the IT company.

Some threats focus on attacking the physical infrastructures of I4.0, such as the attacks in port facilities in Antwerp (Belgium) [88] and the Eli Lilly warehouse [89], where the attackers try to root or steel the products. An attack at the physical infrastructure could also modify the control standards of the production line, such as the attack in the labeled management mode where the meat was labeled with a different type of animal. In 2010, explosive material camouflaged through printer cartridges was found on a cargo plane located at East Midlands Airport [90].

## 6. Discussion—Conclusions

This work presents the main threats for targeting the potentials and the technology assets of I4.0. All industrial organizations must be aware of the existing threats involved in I4.0 ecosystem assets and define a risk assessment methodology for addressing them. This novel industrial environment involves many human actors and processes for improving production procedures by interconnecting various technologies, such as IoT, the cloud, CPS, and AI, and providing on-demand control of the production line. However, this real-time interconnected environment increases the possibility of critical data disclosure [43].

All assets of I4.0 must guarantee the confidentiality and integrity of the information, and privacy and anonymity, especially for industrial users. Authentication and encryption are vital to I4.0 for protecting the data transmitted in different stages and assets and stored in shared sources and preserving the privacy of data owners. Finally, it is crucial to guarantee all the services, users' actions, and produced products.

To conclude, security is essential in the I4.0 ecosystem, and the following actions have to be taken into consideration:

- Ensure the integrity of shared data: No unauthorized changes should be made on stored or shared sources. Data integrity in I4.0 is achieved through symmetric and PKI cryptographic signing schemes. Moreover, hash functions can be applied for archiving accountability of the produced data.
- Secure the communication: Guarantee the confidentiality of the physical/digital information managed and/or produced in the I4.0 environment.
- Ensure continuous certification of actors: Traditional authentication mechanisms, such as symmetric or PKI cryptographic, can guarantee the confidentiality and integrity requirement, but they cannot preserve the user's privacy for increasing the trustworthiness between the participated actors. Privacy attribute-based credentials (P-ABCs) [91] allow users to disclose certified information, minimally authenticating with online service providers. There are several attempts ([92–95]) to use PETs technology to provide an identity-based management scheme via internet providers, smartphones, and the cloud, but it does not apply to all the I4.0 systems' actors such as PLCs and IoT devices. Confidentiality requirements for the communication between IoT devices, PLCs, and servers can be achieved by applying symmetric encryption schemes or PKI cryptographic tools. Privacy ABCs can provide an identity management scheme for authenticating the actions of all actors of I4.0 for securing I4.0 infrastructure by providing an access control system and data sharing policies. PET technology could be used for utilizing centralized identity management schemes for providing trust mechanisms. Blockchain technology [96] and differential privacy techniques [97] could also preserve user's privacy by providing a distributed trust management scheme.
- Security robustness to faults/malicious events: Sensors and industrial equipment are typically prone to faults (as a result of the low-cost equipment), the various conditions of deployment, and may be forced to faults by an attacker. In all cases, P-ABC identity management could revoke the device's credentials, but a monitoring/detection mechanism is needed to verify the availability of the I4.0 assets. Unfortunately, we have to deal with the fact that a component of I4.0 may be compromised, and the industrial company will not dispose of the incident thinking of its reputation. In this case, we must preserve the integrity of any component's services and constantly exploit the compromised component. In order to meet robustness requirements, any asset of the I4.0 should be monitored by a detection mechanism that must be combined with IoT/CPS components for creating a continuously monitoring, secure industrial environment. Recent research on blockchain technology introduced efficient distributed detection mechanisms and risk management schemes [98–100].
- Increase user awareness of security tools/features: All the actors should be aware of the I4.0 security vulnerabilities and risks. A lack of industrial employees' cybersecurity awareness concerning information on attacks and vulnerabilities and a lack of cybersecurity training industrial personnel and industrial stakeholders will increase

potential risks for the industry. Methodologies for training users on security tools and features of I4.0 infrastructure are crucial for building a secure and trustworthy I4.0 ecosystem [62,101].

Although the existing security tools are not mainly targeting handling the presented threats of the I4.0 ecosystem, we presented security solutions for actors to perform their operations securely and privately to decrease the vulnerabilities. One major challenge for making the advent of I4.0 promising and with real potential is to address the threats mentioned above by developing security solutions. It is thoughtless to promise that our suggestions and tools could provide an end-to-end security solution in the interconnected heterogeneous and dynamic ecosystem of I4.0. However, by implementing continuous certification, all the actors of I4.0 can increase the trustworthiness between them. The above cybersecurity solutions can help I4.0 users guarantee the integrity and confidentiality of the produced information and protect users' privacy. Within this collaborative environment, all actors will use various tools to facilitate the secure and private exchange of information. A specific risk management scheme is mandatory for monitoring and detecting anomalies and security violations.

Indeed, we recognize that the field of I4.0 is vast and cannot fully be studied. We attempted to thoroughly examine several crucial aspects to the best of our knowledge. Furthermore, it is beyond doubt that I4.0 is a rapidly evolving domain; thus, it is reasonable to face an obstacle in fully covering all the latest progress. Here we tried to propose security solutions for decreasing the presented vulnerabilities of I4.0. However, if we want to maintain the safety and security levels of I4.0, we must use different means and mechanisms, such as resilience mechanisms, regulatory frameworks, exhaustive validation processes, auditing, and accountability. Moreover, the presented security solutions can not address all potential security challenges of an evolving I4.0 ecosystem; thus, our recommendations have to be combined with various mechanisms in order to build a secure and sustainable ecosystem for I4.0, such as risk assessment schemes, event management mechanisms for prevention and detection, assurance measures, methodologies for compliance with regulatory frameworks, and standardization and certification measures.

We have demonstrated the diversity and nature of several I4.0 state-of-the-art vital components. However, to serve their role complying with all the security standards, they have to address all the security issues. The integration of a plethora of technologies and elements, despite the provision of multiple interconnected services and the improvement of the production line, forms a complicated interconnected environment where threat management constitutes a challenging mission. The solution to several security issues that ultimately emerge is not something to be underestimated. Our work focuses on investigating the specific factors that can harm an I4.0 subsystem. We believe that the categorization of the potential threats is a vital asset of this work. We examined the factors that should be taken into account and could severely harm an I4.0 integrated system, the elements of this system that can affect, and which are the security properties that are related with, in an attempt to answer the research question initially stated as: Which vulnerable elements should be considered in designing and developing an I4.0 network, and how are they connected to the components of I4.0 and the technology provided?

Therefore, we presented the threats that can make each I4.0 element vulnerable. We proceeded with categorizing the threats based on the goals and the elements of I4.0 that they affect and the various cyber-attacks that have successfully occurred and prove their impact on the proper function of I4.0. We believe that the study provided will ultimately help the designers, the ones implementing systems based on the technologies of I4.0, those examining the flaws of a system, and those trying to shield the backdoors of an I4.0 system. Hence, we consider that this work is crucial, mainly targeting the detection and management of I4.0 security risks, security hardening of I4.0 infrastructures (including cyber and physical systems), and risk management methodologies and frameworks.

**Author Contributions:** Conceptualization, C.S. and V.L.; methodology, C.S.; validation, V.L., C.S., and L.P.; investigation, L.P.; resources, L.P. and V.L.; writing—original draft preparation, L.P. and V.L.; writing—review and editing, C.S. and A.P.; visualization, L.P.; supervision, A.P.; project administration, A.P.; funding acquisition, C.S. All authors have read and agreed to the published version of the manuscript.

**Funding:** This study is supported by the project "embeddiNg kEts and Work based learning into MEchaTROnic profile (NEW METRO)" number 600984-EPP-1-2018-1-IT-EPPKA2-SSA co-funded by the Erasmus+ Programme of the European Union and by project "Practical Learning of Artificial iNtelligence on the Edge for indusTry 4.0 (PLANET4) number 621639-EPP-1-2020-1-IT-EPPKA2-KA co-funded by the Erasmus+ Programme of the European Union and by the project "Immersive Virtual, Augmented and Mixed Reality Center of Epirus" (MIS 5047221) implemented under the Action "Reinforcement of the Research and Innovation Infrastructure", funded by the Operational Programme "Competitiveness, Entrepreneurship and Innovation" (NSRF 2014-2020) and co-financed by Greece and the European Union (European Regional Development Fund).

**Conflicts of Interest:** All authors declare no conflict of interest.

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
