# Peer review of "Challenges and Opportunities in Industry 4.0 for Mechatronics, Artificial Intelligence and Cybernetics"

_electronics, doi:10.3390/electronics10162001_

Round 1

Reviewer 1 Report

The authors have addressed most of my comments/suggestions. However, some minor typos remain. For example, in Table 2 you write "threads" several times, I assume that this is supposed to be "threats." So I suggest another round of proofreading. 

Author Response

Thank you for your comment. You are right, Yes,  we mean “threats”. An exhaustive proofreading of the manucript was also  conducted.

Reviewer 2 Report

I read with interest this manuscript, which differs from other review papers in the solid methodological framework that the authors proposed. I provide below some suggestions to improve the article and make it suitable for publication.

  • The research question would be better placed at the end of paragraph 1.Introduction and should be based on a literature gap that the authors should well highlight. In contrast, in the abstract gap and research question would only want to be anticipated to stimulate the interest of the potential reader.
  • The paragraph 2.Main Contribution should be moved to the conclusion paragraph where the authors should highlight the contributions of the study. In the current position the findings are anticipated.
  • Paragraphs 3, 4 and 5 of conceptual narrative, while including the key issues related to the Industry 4.0 paradigm, does not include an issue considered important by both scholars and practitioners i.e. the environmental or more generally sustainability topic. This is shown by the great interest given in the literature to the relationship between Industry 4.0 and environmental issues. This aspect should be included by the authors in the manuscript by referring to the most recent literature on this topic, for example:

https://scholar.google.com/scholar?hl=en&as_sdt=0%2C5&as_ylo=2020&q=%22Industry+4.0+environment%22+and+sustainability&oq=%22Industry+4.0+environment%22+and+sustaina

  • In section 6.Discussion-Conclusion the authors should recall the research question formulated in the introduction and argue that the results provided comprehensive answers to this question. They should also discuss the theoretical and managerial implications of their findings and mention the limitations of this study.

Author Response

The research question would be better placed at the end of paragraph 1.Introduction and should be based on a literature gap that the authors should well highlight. In contrast, in the abstract gap and research question would only want to be anticipated to stimulate the interest of the potential reader.

Answer:

Thank you for your comments, based on your suggestion, we have removed the research question from the abstract and a new paragraph containing the research question has been added at the end of the Introduction (lines 74-83).

The paragraph 2.Main Contribution should be moved to the conclusion paragraph where the authors should highlight the contributions of the study. In the current position the findings are anticipated.

Answer:

We chose to maintain the main contribution section at the initial place of the manuscript because it helps to understand the direction of our work. But, following your suggestion, we restated the  contribution of the paper in the conclusion (lines 904-926).

Paragraphs 3, 4 and 5 of conceptual narrative, while including the key issues related to the Industry 4.0 paradigm, does not include an issue considered important by both scholars and practitioners i.e. the environmental or more generally sustainability topic. This is shown by the great interest given in the literature to the relationship between Industry 4.0 and environmental issues. This aspect should be included by the authors in the manuscript by referring to the most recent literature on this topic, for example:

https://scholar.google.com/scholar?hl=en&as_sdt=0%2C5&as_ylo=2020&q=%22Industry+4.0+environment%22+and+sustainability&oq=%22Industry+4.0+environment%22+and+sustaina

Answer:

Based on your suggestions, we studied this interesting issue and other relative and we updated our manuscript, you may find them at lines 183-190.

In section 6.Discussion-Conclusion the authors should recall the research question formulated in the introduction and argue that the results provided comprehensive answers to this question. They should also discuss the theoretical and managerial implications of their findings and mention the limitations of this study.

Answer:

Following your suggestion, in the conclusion part, we have recall the research question and we have provided comprehensive answers, the updates section is at lines 904-926. Concerning your suggestion to discuss the theoretical and managerial implication of our findings, we have presented them at lines 890-903.

Round 2

Reviewer 2 Report

Dear Authors 

I have read the latest version of your manuscript appreciating the improvements you have made by following the reviewers' recommendations. Therefore, I now consider your article suitable for publication in this journal.

Congratulations!